# The Antimicrobial Potential of the Hop (*Humulus lupulus* L.) Extract against *Staphylococcus aureus* and Oral *Streptococci*

**DOI:** 10.3390/ph17020162

**Published:** 2024-01-27

**Authors:** Alyona Khaliullina, Alyona Kolesnikova, Leysan Khairullina, Olga Morgatskaya, Dilyara Shakirova, Sergey Patov, Polina Nekrasova, Mikhail Bogachev, Vladimir Kurkin, Elena Trizna, Airat Kayumov

**Affiliations:** 1Institute of Fundamental Medicine and Biology, Kazan Federal University, 420008 Kazan, Russia; anela_90@mail.ru (A.K.); lagoon_1998@mail.ru (A.K.); aliullina98@mail.ru (L.K.); ol-morgatskaya@yandex.ru (O.M.); dhabilevna@mail.ru (D.S.); trizna91@mail.ru (E.T.); 2Institute of Chemistry, FRC “Komi Scientific Centre”, Ural Branch of the Russian Academy of Sciences, 167000 Syktyvkar, Russia; ser-patov@yandex.ru (S.P.); polina.nekrasova.98@bk.ru (P.N.); 3Biomedical Engineering Research Centre, St. Petersburg Electrotechnical University, 5 Professor Popov Street, 197022 St. Petersburg, Russia; rogex@yandex.com; 4Institute of Pharmacy, Samara State Medical University, 443079 Samara, Russia; kurkinvladimir@yandex.ru

**Keywords:** oral pathogens, antimicrobial resistance, antimicrobials, phytoextracts, synergy

## Abstract

Plant extracts are in the focus of the pharmaceutical industry as potential antimicrobials for oral care due to their high antimicrobial activity coupled with low production costs and safety for eukaryotic cells. Here, we show that the extract from Hop (*Humulus lupulus* L.) exhibits antimicrobial activity against *Staphylococcus aureus* and *Streptococci* in both planktonic and biofilm-embedded forms. An extract was prepared by acetone extraction from hop infructescences, followed by purification and solubilization of the remaining fraction in ethanol. The effect of the extract on *S. aureus* (MSSA and MRSA) was comparable with the reference antibiotics (amikacin, ciprofloxacin, and ceftriaxone) and did not depend on the bacterial resistance to methicillin. The extract also demonstrated synergy with amikacin on six *S. aureus* clinical isolates, on four of six isolates with ciprofloxacin, and on three of six isolates with ceftriaxone. On various *Streptococci*, while demonstrating lower antimicrobial activity, an extract exhibited a considerable synergistic effect in combination with two of three of these antibiotics, decreasing their MIC up to 512-fold. Moreover, the extract was able to penetrate *S. aureus* and *S. mutans* biofilms, leading to almost complete bacterial death within them. The thin-layer chromatography and LC-MS of the extract revealed the presence of prenylated flavonoids (2′,4′,6′,4-tetrahydroxy-3′-geranylchalcone) and acylphloroglucides (cohumulone, colupulone, humulone, and lupulone), apparently responsible for the observed antimicrobial activity and ability to increase the efficiency of antibiotics. Taken together, these data suggest an extract from *H. lupulus* as a promising antimicrobial agent for use both as a solely antiseptic and to potentiate conventional antimicrobials.

## 1. Introduction

The human oral cavity is a habitat for many microorganisms, including bacteria, archaea, fungi, viruses, and protozoa. Various internal and external environmental factors affect the oral microbiota, leading to dynamic changes in its composition, from insignificant alteration to dysbiosis. Consequently, various diseases develop, such as caries, periodontal disease, oral candidiasis, and systemic infections [1,2,3]. Furthermore, biofilm formation significantly complicates the treatment of oral infections with common antimicrobials. Due to the inability of antimicrobials to penetrate the extracellular matrix, the effect of antimicrobials on bacteria in the biofilm is significantly lower compared to free-floating planktonic cells. Therefore, the development of new antiseptics with low toxicity for the mucosa and a low risk of resistance development that are thus acceptable for daily use, while being efficient against the oral pathogenic flora, including biofilms, is challenging.

The primary colonizers of the oral cavity are *Streptococcus sanguinis*, *S. oralis*, *S. intermedius*, *S. gordonii*, *Peptostreptococcus micros*, *Gemella morbillorum*, and *Actinomyces* species [4,5,6]. *Streptococci* and Actinomyces colonize the oral mucosa of a baby from the first days of life and come from the microbiota of the mother [7]. However, over the lifecycle, environmental pressure leads to an imbalance of microorganisms or an abundance of pathogenic ones [8,9,10]. As a consequence, dramatic changes in the production of acid and alkali by bacteria occur, which, in turn, lead to tooth surface damage and caries development. The development of caries is closely related to the formation of polymicrobial biofilms by the oral microbiota on teeth and soft tissues [11]. *Lactobacilli* and *Streptococcus mutans* play a main role in the occurrence and progression of caries due to the fermentation of sucrose and other sugars with the formation of lactic acid, which plays a major role in the local acidification of the caries environment [12,13,14,15,16]. In addition, the symbiotic interaction of *Porphyromonas gingivalis* with *S. gordonii* or *S. oralis* has been shown to lead to significant complications in the development of disease and bone loss [17,18]. In addition to *Streptococci*, *Staphyloococcus aureus* is one of the main microorganisms involved in the development of periodontitis [19]. Thus, *S. aureus* forms mixed biofilms with *Streptococci* and causes inflammation [20,21]. Therefore, the search for substances that can penetrate through the biofilm and have a bactericidal effect on bacteria embedded in the extracellular matrix is an important challenge.

Phytoextracts are widely known for their therapeutic effects due to their high antimicrobial and antibiofilm activity [22,23,24]. Their natural origin, low toxicity, and low economic costs make them attractive for pharmaceuticals [25,26]. Moreover, the use of chemicals from phytoextracts as antimicrobials has a low probability of bacterial resistance development [27,28,29]. The beneficial use of phytoextracts for antimicrobial treatment is also due to the synergistic effects of active components in their composition. Synergism is provided by various effects, like the multitargency of substances, the ability to suppress intracellular resistance mechanisms, and the increased bioavailability of substances [30]. Thus, the use of bearberry and cranberry juice contributed to the effective treatment of urinary tract infections. Extracts of lemon balm (*Melissa officinalis*), garlic (*Allium sativum*), and tea tree (*Melaleuca alternifolia*) have been described as broad-spectrum antimicrobial agents [31]. Also, high antibacterial activity against *S. aureus*, *Escherichia coli*, *Pseudomonas aeruginosa*, and *Salmonella typhimurium* has been described for *Myrtus communis* and *Verbena officinalis* [32]. Extracts of the bark and leaves of *Eucalyptus camaldulensis* showed high antimicrobial activity against a wide range of Gram-positive and Gram-negative bacteria, while the effective concentration varied from 0.08 μg/mL to 200 μg/mL and depended on the extraction procedure [33]. In vitro studies have shown that catechins found in green tea extracts are able to inhibit the growth of *Vibrio cholerae*, *S. mutans*, and *Shigella* [34,35,36]. Soursop leaf extract [37] had antimicrobial activity similar to that of chlorhexidine against the most significant cariogenic microorganisms, such as *S. mutans*, *S. mitis*, and *C. albicans*. *In vivo* studies of garlic extract have shown antimicrobial activity against pathogenic *Streptococci* [38,39,40,41].

In addition to their antimicrobial activity, phytoextracts are also able to increase the effectiveness of systemic antibiotics. Thus, mangosteen extract increased the effectiveness of β-lactam antibiotics against several resistant strains [42]. *Camellia sinensis* dry leaf extract increased the effectiveness of nalidixic acid against *Salmonella typhi* by eight times [43]. Also, the combination of individual components of extracts from *Jatropha elliptica* demonstrated a synergistic effect with fluoroquinolones against *S. aureus* [44]. The use of carnosic acid isolated from *Rosmarinus officinalis* L. extract potentiated the effect of erythromycin against multidrug-resistant bacteria [45]. Myrtenol was able to increase the efficiency of various antimicrobials against *S. aureus* and *C. albicans* [46].

The antimicrobial properties of hops (*Humulus lupulus* L.) have been well known for a long time and have been used for beer production since the 11^th^ century, mainly to repress bacteria that spoil the beer [47]. Since then, the antimicrobial activity of hop infructescence extracts against a wide range of bacteria, as well as its antiviral and antifungal activity, has also been reported [48,49]. Furthermore, hop extracts do not have a toxic effect on the human body [50,51,52], which makes them suitable for use as an antiseptic. The antimicrobial activity of hop essential oil has been also shown against a number of pathogens, such as *Yersinia enterocolitica*, *Salmonella enteritidis*, *S. typhimurium*, *Proteus mirabilis*, *Escherichia coli*, *Klebsiella oxytoca Enterobacteriaceae*, *Enterococci*, and anaerobic bacteria [48,53,54,55]. However, only a small number of studies have been devoted to evaluating its effect on oral microflora, both alone and in combination with other antimicrobials [56].

Here, we report that the ethanol extract from hop infructescence demonstrates high antimicrobial activity and a synergetic effect with various conventional antimicrobials against *S. aureus* (both MSSA and MRSA), *S. gordonii*, *S. mutans*, *S. sobrinus*, and *S. salivarius*, including the biofilm-embedded forms, thus being an attractive phytosubstance for oral care.

## 2. Results

### 2.1. Metabolite Profile of Hop Extracts

In the first step, the extraction of the active compounds from hops was performed by the reflux technique using an eluotropic variety of solvents: hexane, chloroform, acetone, 95% ethanol, and 70% ethanol. The extracts obtained were then separated with TLC, and the extraction efficiency was evaluated by the yield of hop flavonoids with each solvent (Figure 1, lanes 4–8) in comparison with luteolin, rutin, quercetin, and xanthohumol (Figure 1, lanes 1–3 and 10, respectively). All extracts seem to contain xanthohumol and quercetin. Xanthohumol, the prenylated flavonoid, is located on the top of the TLC plate, higher than other less polar flavonoids, and appears as a spot with dark-green fluorescence. Furthermore, the highest yield of total flavonoids was observed after acetone extraction (lane 4); therefore, this solvent was used for further preparation of extracts on a preparative scale.

The final extract obtained by acetone extraction appeared as a dark, viscous orange liquid. It was dissolved in 95% ethanol at the final concentration of 40 mg/mL and used for other tests as a pharmaceutical substance from *H. lupulus*. The TLC analysis of the extract (Figure 1, lane 9) revealed the presence of xanthohumol, while other polar flavonoids were not detected. Furthermore, the Beer–Lambert law-based UV-vis-spectrophotometric qualitative analysis of the obtained extract revealed the maximum optical absorbance at λ = 285 nm and λ = 334 nm (Appendix A) that correspond to the characteristic absorption lines of APs.

For a deeper analysis of the compounds in the extract, LC-MS was performed. Among the 10 most abundant compounds identified with LC-MS (see Appendix A for chromatogram), prenylated flavonoids (2′,4′,6′,4-tetrahydroxy-3′-geranylchalcone) and acylphloroglucides (cohumulone, colupulone, humulone, lupulone, and others) were detected (see Table 1 and Appendix A for structures), suggesting potential antimicrobial activity of the obtained substance.

### 2.2. Antimicrobial Effect of an Extract from H. lupulus

The antimicrobial effect of an extract from *H. lupulus* was tested on methicillin-sensitive *S. aureus* ATCC 29213 and six clinical isolates of *S. aureus* (three MSSA and three MRSA), as well as on four different strains of *Streptococci* (*S. mutans*, *S. sobrinus*, *S. salivarius*, and *S. gordonii*) (Table 2). The fraction of *H. lupulus* infructescence demonstrated high antimicrobial activity against all studied *S. aureus* strains. The minimum inhibitory concentration (MIC) varied within 10–40 µg/mL of total compounds in the extract and did not depend on bacterial sensitivity to methicillin. At the same time, the effective concentrations of the extract in relation to individual strains were comparable to or significantly lower than the MIC of the reference antibiotics, excepting *S. aureus* 68 isolate, which was sensitive to all antimicrobials (Table 2). On the contrary, the extract demonstrated moderate activity against *S. mutans*, *S. sobrinus*, *S. salivarius*, and *S. gordonii* (MIC for all *Streptococci* was 625 µg/mL) compared to reference antibiotics.

### 2.3. Potentiation of Antimicrobial Agents by the H. lupulus Extract

The combined effect of the hop extract and antibiotics was evaluated by the checkerboard assay. For both *S. aureus* and *Streptococci*, the final concentration of the studied antimicrobials was in the range of 0.06–2×MIC. After 24 h of incubation, the fractional inhibitory concentration index was calculated.

The combined use of an extract and amikacin increased the efficiency of the antibiotic against four of six *S. aureus* strains by 8–1024-fold, suggesting considerable synergy (Table 3). The synergism was less pronounced when the extract was combined with ciprofloxacin and ceftriaxone. In relation to *S. aureus* ATCC 29213, the combination of these antimicrobials with an extract from the hop demonstrated an additive effect with FICI of 0.75 and 1.25, respectively. Nevertheless, for half of clinical isolates, in the presence of extract, the bacterial susceptibility to antimicrobials increased up to 128-fold; however, this effect was strain-specific and did not depend on resistance to methicillin (Table 3). Of note, an extract was unable to increase the bacterial susceptibility to those antimicrobials that were inefficient for the given isolate, apparently because of their genetically determined resistance.

Next, the synergism of the hop extract and antibiotics was evaluated on four different isolates of *Streptococci*. Despite the low antimicrobial activity on these bacteria, the combined use of the extract with antibiotics increased the efficiency of the latter considerably. Thus, the MIC of amikacin and ceftriaxone in the presence of the extract against *S. mutans* decreased 64- and 8-fold, respectively, while the combination with ciprofloxacin showed an additive effect (Table 3). In relation to *S. sobrinus* and *S. salivarius*, the MIC of all three antibiotics (except amikacin for *S. salivarius*) in the presence of hop extract decreased 256–512-fold (FICI = 0.25). However, in relation to *S. gordonii*, the synergism of the extract was observed only when it was combined with amikacin (FICI = 0.26), while for ciprofloxacin and ceftriaxone, either antagonism (FICI = 8.6) or an additive effect (FICI = 1.25) was detected, respectively (Table 3).

Thus, these data clearly show that the combined use of the extract from hop infructescence is capable of significantly increasing the efficiency of antimicrobial therapy for the most significant pathogenic microorganisms of the oral cavity, including those with antimicrobial resistance.

### 2.4. Extract from H. lupulus Kills Bacterial Cells in Biofilm

It has been reported in several works that extracts of medicinal plants are able to target biofilm-embedded bacteria, which makes them attractive agents for oral care [57]. We tested the effect of the extract from *H. lupulus* infructescence on mature biofilms of *S. aureus* (MSSA and MRSA), *S. mutans*, and *S. sobrinus*, common pathogens causing oral diseases. A 0.05% solution of chlorhexidine, an antiseptic widely used as a mouthwash [58], was used as a reference. For that, 48-hour-old biofilms were treated for 3 h and then analyzed with confocal laser scanning microscopy (Figure 2). CLSM data showed that the treatment of *S. aureus* biofilms, formed by both MSSA and MRSA isolates with an extract solution at a concentration equal to 2×MIC, led to the death of almost all cells after 3 h of treatment. The fraction of viable bacteria in the presence of hop extract was 9–15%, while the treatment of biofilms with chlorhexidine caused the death of only 55–56% of cells (Figure 2). Also, the extract was effective against *S. mutans* cells in the biofilm since the average fraction of viable cells in the biofilm was 11%, while the treatment with chlorhexidine led to the death of 24% of the cells. On the contrary, the cells in the *S. sobrinus* biofilm were not sensitive to extract under the conditions tested (the fraction of viable cells consisted 82% of the entire biofilm), whereas chlorhexidine treatment led to 61% of residual viability (Figure 2).

A deeper analysis of the distribution of viable cells in biofilms revealed that an extract has worse penetration into the *S. aureus* biofilms, and a significant decrease in viable cells (more than twice) was already observed in the middle layers of the biofilm. By contrast, the extract was able to penetrate the *S. mutans* biofilm, which was confirmed by only a few viable cells in the bottom layers. At the same time, the *S. sobrinus* biofilm was the least permeable for an extract, and the number of viable cells throughout the entire biofilm layers varied between 70 and 80% (Figure 2). In turn, chlorhexidine penetrated the biofilms of all studied strains. However, only *S. aureus* MSSA cells were sensitive to the antiseptic, where almost complete cell death could be observed in the bottom layers of the biofilm. At the same time, in the biofilms of MRSA and both *Streptococci*, cell viability in the presence of chlorhexidine in the bottom layers varied within 40–50%, which confirms the low efficiency of this antimicrobial against biofilms (Figure 2).

## 3. Discussion

Dental caries and periodontal diseases are considered the most common chronic diseases of the oral cavity worldwide, and currently, there are only a few approved antimicrobials for their treatment [8]. To date, chlorhexidine remains the most prevalent treatment agent widely used in oral care due to its broad spectrum of antimicrobial activity and prolonged action. However, the prolonged use of chlorhexidine in sublethal concentrations induces resistance development by oral bacteria [59] and has various side effects [58,60]. Plant extracts with antimicrobial properties seem like a promising alternative to synthetic antibacterial agents, and the search for new sources of plant-derived antimicrobial agents that are generally non-toxic and do not induce resistance remains an urgent task to enhance the efficiency of infectious disease treatment.

In this study, we investigated the antimicrobial activity of an extract from *H. lupulus* against some oral pathogens. A specific component of hop extracts, a bitter acid xanthohumol, has been reported as targeting the membrane of Gram-positive bacteria and thus exhibiting high antimicrobial activity [56,61,62], and its presence in the extract has been confirmed by TLC (Figure 1). Indeed, the hop extract demonstrated significant antimicrobial activity against *S. aureus*, and this activity was not dependent on the bacterial resistance to methicillin (Table 2), that fits with other studies [63]. By contrast, the activity of the extract from hop was low on *Streptococci* (*S. mutans*, *S. sobrinus*, *S. salivarius*, and *S. gordonii*). This may be associated with the ability of *Streptococci* to alter the acid-base balance of the environment during their lifecycle [64,65,66], and the effectiveness of hop extracts has been reported to be directly dependent on the pH of the medium. Nevertheless, the presence of antimicrobial activity against both *S. aureus* and various strains of *Streptococci* opens possibilities for treatment of mixed infections, which are dominant in periodontal diseases [67].

Bacteria associated with the development of dental caries and periodontal diseases are often present in the form of polymicrobial communities [19]. Therefore, the development of a universal antiseptic agent active against a wide range of bacteria embedded in biofilms remains an urgent task for pharmaceuticals. Extract from *H. lupulus* demonstrated the ability to penetrate into *S. aureus* (MSSA and MRSA) and *S. mutans* biofilms, which fits with earlier data [68]. Thus, the fraction of viable cells in treated biofilms decreased up to 9–15%, including the bottom layers of the biofilm (see viable cell distribution on Figure 2). At the same time, the effect of chlorhexidine was less pronounced, especially against *S. mutans* biofilm, which seems to be a consequence of the low permeability of the antiseptic through the biofilm matrix. Surprisingly, the cells in the *S. sobrinus* biofilm were insensitive to both chlorhexidine and the extract, apparently because of the low permeability of *S. sobrinus* biofilms to antimicrobials: the fraction of viable cells in all layers of the biofilm varied in the range of 60–90% after 3 h of treatment with either chlorhexidine or the extract. The extracellular matrix of biofilms of different bacteria differs in biochemical composition as well as physico-chemical properties, which could affect the biofilm’s permeability to antimicrobial agents and their activity [69,70,71]. Nevertheless, the high antimicrobial activity of the hop extract against biofilm-embedded cells of three of the four studied strains suggests its potential as a promising antibiofilm agent.

Currently, to overcome microbial resistance to antimicrobial agents, complex therapy with several compounds is used. For that, essential oils, phytoextracts, and active antimicrobial components of medicinal plants are increasingly considered as antibiotic enhancers for both susceptible and resistant microorganisms [42,72]. The hop extract demonstrated high synergistic potential with antibiotics from three different groups used as model ones. Thus, the efficiency of amikacin increased against five out of the seven *S. aureus* strains used in this study, with a 1024-fold increase for *S. aureus* 73 (Table 3). Additionally, the introduction of the extract increased the susceptibility of 67% and 50% of *S. aureus* clinical isolates to ciprofloxacin and ceftriaxone, respectively (Table 3). The combination of the extract with antimicrobials also led to an increase in the effectiveness of the latter against *Streptococci*. A synergetic effect on three out of four strains (*S. mutans*, *S. sobrinus*, and *S. gordonii)* was shown when the extract was combined with amikacin. In the case of ciprofloxacin and ceftriaxone, the synergism of the extract was shown in relation to two (*S. sobrinus*, *S. salivarius*) and three (*S. mutans*, *S. sobrinus*, *S. salivarius*) species of *Streptococci* out of four, respectively (Table 3). Natarajan et al. showed in their study that the synergistic effect of the active components of hops and synthetic antibiotics (polymyxin B, ciprofloxacin, and tobramycin) does not depend on the structure of the bacterial cell wall [73]. It is probable that the different degree of synergy between antibiotics and the extract of hops used in our study may have been due to the physicochemical interactions of molecules, as well as the different degree of sensitivity of bacteria to the components of the extract. For example, a comparative evaluation of the synergism of the crude hop extract and xanthohumol with cefepime, ceftriaxone, ciprofloxacin, and sparfloxacin showed a synergistic or additive effect against Gram-positive bacteria when antibiotics were combined with the crude extract, while xanthohumol did not show similar results [64]. Thus, it is likely that the synergy of the extract of hops is provided by other secondary metabolites present in the phytoextract, and the leading compound remains to be elucidated.

Taken together, our data allow us to suggest the hop extract as either a solely antibacterial substance or an enhancer of various antimicrobials for oral care. It could be speculated that the observed synergistic effect of the extract with aminoglycoside, fluroquinolone, and cephalosporin antibiotics could be an indicator of a putatively similar effect with other classes of antimicrobials for systemic and topical applications. The increase in their efficiency would make it possible to decrease the required concentration of antimicrobials and thus reduce the risk of both side effects and bacterial resistance development.

## 4. Materials and Methods

### 4.1. Plant Material and Sample Preparation

The raw plant material (hops of *H. lupulus* from the *Cannabaceae* family) was harvested in Yutazy district of the Republic of Tatarstan (54°35′28.0″ N 53°16′52.5″ E) on 17 August 2022, in accordance with World Health Organization (WHO) guidelines on good agricultural and collection practices (GACP) for medicinal plants. The hops were dried at 25 °C and used for the extraction of common bioactive compounds (BAC): acylphloroglucides (APs) and prenylated flavonoids (PFs). The samples of hops corresponded to the requirements of official pharmaceutical documents (European Pharmacopoeia, 11th edition) on morphological and microscopical characteristics. The hop cones were milled using the laboratory grain mill LGM-1 (OLIS LLC, Moscow, Russia) until the particle size became less than 2 mm. The analytical samples for extraction contained crushed hops and lupulin glands.

### 4.2. Solvents, Chemicals, and Apparatus

Acetone and hydrochloric acid were analytical grade and purchased from JSC «EKOS-1» (Moscow, Russia). The pure deionized water was prepared using an automatic purification system (Merck Millipore, Darmstadt, Germany). Xanthohumol (purity ≥ 96% for HPLC) was purchased from Merck (Darmstadt, Germany), and its solution in ethanol (final concentration of 0.1 mg/mL) was prepared as a reference. Other reference standards for flavonoids (rutin, quercetin, and luteolin) were commercial standards from the Institute of Pharmacy of Samara State Medical University, which were authenticated by standard analytical chemistry techniques (_1_H, _13_C NMR). Solutions of rutin, quercetin, and luteolin had a final concentration of 5 mg/mL and were stored at 4 °C, protected from light. Amikacin, ciprofloxacin, and ceftriaxone were purchased from Sigma (St. Louis, MO, USA). Stock solutions were prepared at a concentration of 20 mg/mL in deionized water.

### 4.3. Extraction, Isolation, and Purification of Bioactive Compounds from Hop

On the first step, hop extracts were obtained using various solvents (acetone, hexane, 95% ethanol, 70% ethanol, and chloroform) using the reflux extraction technique. Briefly, milled hop infructescence (1 g) was subjected to 15 min of reflux at boiling temperature of the solvent (10 mL) on Biosan-Grant SUB Aqua Pro, Royston, UK.

As we learned from our investigations, the best extraction efficiency has been observed when using acetone. Therefore, the next extraction procedures were performed with pure acetone. The bioactive compounds from *H. lupulus* were isolated and purified using preparative methods described in [74] with modifications. Briefly, the dry milled hop cones (10 g) were mixed with 150 mL of acetone and incubated at room temperature for 1 h with stirring at 200 rpm. An extract was filtered through a paper filter (5 µm), and one volume of pure water was added. Then, the pH of the mixture was adjusted to 6.0 with 1% HCl, and the solution was concentrated on a rotary evaporator (Rotavapor^®^ R-300, Lausanne, Switzerland) at a temperature not exceeding 56 °C with a vacuum of 5 mbar and a stirring speed of about 100 rpm. The residual organic fraction was kept.

### 4.4. Thin-Layer Chromatography

Thin-layer chromatography (TLC) was performed on TLC plates with a size of 10 × 15 cm and a thickness of about 90–120 µm (Aluminum TLC Silica Gel Sorbfil, Krasnodar, Russia). Before TLC, a TLC chamber was pre-equilibrated for 30 min with solvent vapors by lining the chamber with filter paper moistened with mobile-phase solution (chloroform-ethanol-water; 25:18:2; *v*/*v*/*v*). Drops of each extract solution and working standard solutions were loaded onto the baseline of the TLC plate by syringe, and the plates were placed into the pre-equilibrated TLC chamber. After separation, TLC plates were dried and observed at 366 nm in a UV Cabinet 4 (CAMAG, Muttenz, Switzerland). The extracted compounds were visualized using derivatization by a 2% solution of aluminum chloride in ethanol, the basic chemical revelator for flavonoids.

### 4.5. LC-MS Analysis

Initially, 1 mL of the final extract was dissolved in 10 mL of pure water and loaded onto Hypersep C18 (Thermo Electron Corporation, 100 mg/1 mL/100 pg, Waltham, MA, USA), and after being washed by 20 mL of pure water, bound compounds were eluted in 1 mL of acetonitrile:pure water (containing 10% formic acid) with a ratio of 1:1 (*v*/*v*). The samples were analyzed by HPLC-MS on a Thermo Finnigan LCQ fleet (Themo Fisher Scientific, Waltham, MA, USA) equipped with a BDS Hypersil C 18 column (2 × 150 mm, 5 µm). Then, 1 µL of the sample was loaded, and the chromatography was performed in an isocratic mode with a flow rate of 0.8 mL/min, in a solvent system of acetonitrile-water (10% formic acid and 0.01% i-propanol) in a ratio of 1:1 (*v*/*v*) and detection on a PDA detector at 335 nm.

The mass spectrometry (ESI) was performed at 275 °C with a capillary voltage of 5 kV in helium flow (7 L/min) and the detection of positive ions (M^+^) in the mass range of 100–2000 (*m*/*z*). Data acquisition was carried out with the Xcalibur data system (Thermo Finnigan, Themo Fisher Scientific, Waltham, MA, USA). The compounds were identified by comparing the obtained ESI/MS spectra (retention time and molecular ion weight *m*/*z*) with those for authentic standards described in [75,76,77,78] and additionally obtained for the pure humulone and lupulone (Merck, Lebanon, NJ, USA).

### 4.6. Bacterial Strains and Growth Conditions

The antimicrobial activity of the extract from *H. lupulus* was tested on *Staphylococcus aureus* ATCC 29213 and six clinical isolates of *S. aureus* (3 MSSA and 3 MRSA) provided by the Pharmaceuticals Research Center of Kazan Federal University. To evaluate the antimicrobial activity of the extract against oral microbiota, isolates of *Streptococcus mutans*, *Streptococcus sobrinus*, *Streptococcus salivarius*, and *Streptococcus gordonii* obtained at Kazan Federal University from tooth plaques of healthy volunteers were used. *S. aureus* strains were cultivated in the LB medium. The LB medium supplemented with FBS (5% *v*/*v*) and glucose (2% *w*/*v*) was used for the maintenance of *Streptococci*. The basal medium (BM) (peptone 7g, MgSO_4_ × 7H_2_O 2.0 g and CaCl_2_ × 2H_2_O 0.05g in 1.0 L of tap water) or supplemented with FBS (5% *v*/*v*) and glucose (2% *w*/*v*) was used to obtain *S. aureus* or *Streptococci* biofilms, respectively [79,80].

### 4.7. Determination of Minimal Inhibitory Concentration (MIC)

The minimum inhibitory concentration (MIC) of antimicrobials was determined by serial 2-fold microdilution in 96-well plates according to the EUCAST rules for antimicrobial susceptibility testing [81] with some modifications. The final concentrations of compounds in wells were 0.05–80 µg/mL for the extract and 0.06–2048 µg/mL for the antimicrobials. The wells were seeded with microbial culture with a final density of 10^6^ CFU/mL in LB medium, and plates were incubated at 37 °C for 24 h under static conditions. The concentration of compounds providing no bacterial growth assessed by the Alamar-blue cell viability test, with 0.1% resazurine (Applichem, Darmstadt, Germany) considered as the MIC.

### 4.8. Assessment of Synergy between Hop Extract and Antimicrobials

To assess the ability of extract to potentiate the activity of antimicrobials, a checkerboard assay was performed as described previously [82]. Each plate contained serial dilutions of extract and various antimicrobials in a checkerboard pattern. One of the antimicrobials was diluted horizontally and the extract vertically on a 96-well plate. The last lines and columns contained only one of the considered compounds to determine their MICs in each experiment. The initial concentration of each of the studied compounds was 2×MIC. All wells contained bacterial cultures with a final density of 10^6^ CFU/mL in LB medium. The plates were incubated at 37 °C for 24 h. The experiments were performed in triplicate, and a growth control without the addition of any antimicrobial agent was included in each plate. After incubation, the optical density OD600 was measured on an Infinite 200 PRO plate spectrophotometer (Tecan, Männedorf, Switzerland), and the fractional inhibitory concentration index (FICI) for each double combination was calculated as described earlier [83]. Interpretation of the obtained FICI values was carried out according to Odds [84].

### 4.9. Biofilm Formation

Bacterial biofilms were obtained by cultivation of bacteria in BM broth in a cell imaging chambered coverslip with 8 wells (Ibidi, Gräfelfing, Germany). The wells were seeded with an 18 h-old culture of bacteria until the final concentration of 10^6^ CFU/mL and grown for 24 h under static conditions at 37 °C.

### 4.10. Confocal Laser Scanning Microscopy

To assess the ability of the extract to kill bacterial cells in biofilms, confocal laser scanning microscopy was performed. For that, bacteria were grown in BM medium in a cell imaging chambered coverslip with 8 wells (Ibidi, Gräfelfing, Germany) under static conditions. After 24 h of incubation, fresh broth supplemented with extract with a final concentration equal to 2 × MIC was added. Chlorhexidine was used as the reference antiseptic. After 3 h of incubation, biofilms were stained for 15 min with 3,3′-Dihexyloxacarbocyanine iodide (Sigma-Aldrich, St. Louis, MO, USA) at a final concentration of 0.02 µg/mL (green fluorescence) and propidium iodide (Sigma-Aldrich, St. Louis, MO, USA) at a final concentration of 3 µg/mL (red fluorescence) to differentiate live and dead cells. CLSM was performed using an Olympus IX83 (Olympus Europa, Hamburg, Germany) inverted microscope supplemented with a STEDYCON ultrawide extension platform.

### 4.11. Data Analysis

All experiments were performed in four independent biological repeats, with three technical repeats in each run. The number of viable cells was estimated as the relative fraction of the green cells among all cells in the overlayed images obtained by confocal laser scanning microscopy. For each sample, Z-stackes were individually uploaded and analyzed by BioFilmAnalyzer software (version 1.2) [85].

## 5. Conclusions

Taken together, our results indicate that the extract from *H. lupulus* can serve as a universal antimicrobial agent and be used as both a solely substance highly active against *S. aureus* (both MSSA and MRSA) and an enhancer of the efficiency of various conventional antimicrobials against oral pathogenic cocci (*S. aureus* and *S. mutans*). Thus, the MIC of amikacin decreased 64-, 512-, and 1024-fold against MSSA and the cariogenic *Streptococci S. sobrinus* and *S. mutans*, respectively, and the 256-fold potentiation of ciprofloxacin and ceftriaxone was observed against *S. sobrinus* and *S. salivarilus.* Moreover, the expressed anti-biofilm activity against *S. aureus* (MSSA and MRSA) and *S. mutans* makes the hop extract a promising active component for dental products, providing treatment and prevention of caries and other periodontal diseases associated with biofilm formation.

## Figures and Tables

**Figure 1 pharmaceuticals-17-00162-f001:**
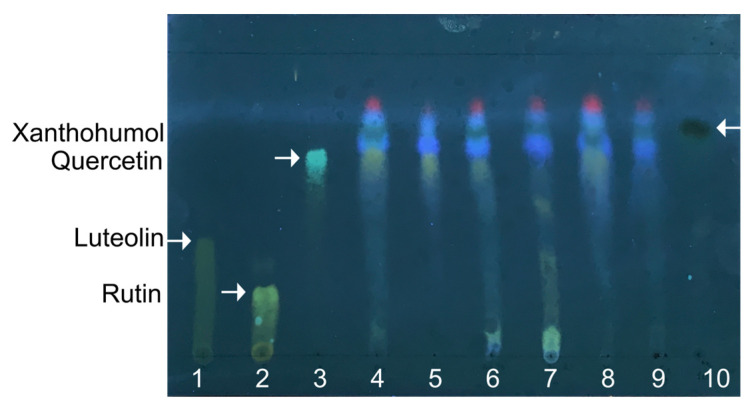
The representative thin-layer chromatography of hops extracts in chloroform, ethanol, and water (25:18:2). Extracts with acetone (4), hexane (5), 95% ethanol (6), 70% ethanol (7), and chloroform (8) were loaded with 5 µL each. Lane (9) corresponds to acetone extract prepared on a preparative scale and dissolved in 95% ethanol. Commercially available pure luteolin (1), rutin (2), quercetin (3), and xanthohumol (10) were used as references, and the corresponding spots are shown by arrows.

**Figure 2 pharmaceuticals-17-00162-f002:**
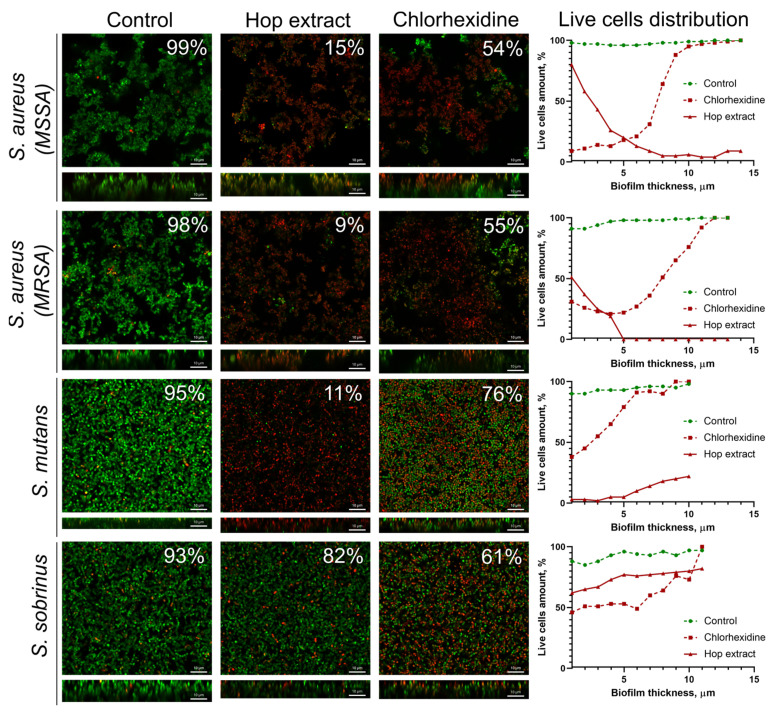
The effect of the hop extract on bacterial cells in biofilms. An extract of *H. lupulus* was added to mature 48 h biofilms of *S. aureus* (MSSA and MRSA), *S. mutans*, and *S. sobrinus* at a concentration equal to respective 2×MIC, incubated for 3 h, and the viability of the bacteria was evaluated using confocal laser scanning microscopy. Living cells are shown in green, and dead cells are shown in red. The fraction of viable cells in each Z-stack was calculated using the BioFilmAnalyzer software (version 1.2) and estimated as the relative fraction of the red cells among all cells in the combined images obtained by overlaying the green and red fluorescence microphotographs. The scale bars indicate 10 μm.

**Table 1 pharmaceuticals-17-00162-t001:** The prenylated flavonoids and acylphloroglucides identified by LC-MS in the hop infructescence extract.

Number	Retention Time	Molecular Weight (*m*/*z*)	Molecular Formula	Identified Compound
1	4.16	407.1	C_25_H_28_O_5_	2′,4′,6′,4-tetrahydroxy-3′-geranylchalcone
2	7.83	333.0	C_19_H_26_O_5_	Posthumulone
3	10.2	347.4	C_20_H_28_O_5_	Cohumulone
4	10.87	408.7	C_25_H_36_O_4_	Colupulone
5	13.71	361.5	C_21_H_30_O_5_	Humulone
6	14.79	423.1	C_26_H_38_O_4_	Lupulone
7	18.09	360.9	C_22_H_32_O_5_	Adhumulone
8	23.94	385.3	C_22_H_32_O_5_	Prehumulone
9	31.85	386.2	C_22_H_32_O_5_	Adprehumulene
10	33.96	446.8	C_27_H_40_O_4_	Adprelupulone

**Table 2 pharmaceuticals-17-00162-t002:** Minimal inhibitory concentrations of the extract from *H. lupulus*, amikacin, ciprofloxacin, and ceftriaxone on bacterial cells. Median values from four biological replicates are shown.

Bacterial Strains and Isolates	MIC, µg/mL			
	Extract from *H. lupulus*	Amikacin	Ciprofloxacin	Ceftriaxone
*S. aureus* ATCC (MSSA)	10	32	16	4
*S. aureus* 18 (MSSA clinical isolate)	10	16	8	2
*S. aureus* 25 (MSSA clinical isolate)	40	16	2048	0.5
*S. aureus* 26 (MSSA clinical isolate)	10	64	8	2048
*S. aureus* 67 (MRSA clinical isolate)	40	2048	8	2048
*S. aureus* 68 (MRSA clinical isolate)	10	4	1	1
*S. aureus* 73 (MRSA clinical isolate)	40	128	2048	1
*S. mutans* (clinical isolate)	625	8	0.06	0.5
*S. sobrinus* (clinical isolate)	625	64	16	16
*S. salivarius* (clinical isolate)	625	2048	16	16
*S. gordonii* (clinical isolate)	625	16	0.06	0.06

**Table 3 pharmaceuticals-17-00162-t003:** Fractional inhibitory concentration and FICI (index of fractional inhibitory concentration) values of antimicrobials in the presence of an extract from *H. lupulus* on bacterial cells. Median values of FICI from four biological replicates are shown.

Bacterial Strains and Isolates	Amikacin		Ciprofloxacin		Ceftriaxone	
	FICI	MIC Decrease, Fold	FICI	MIC Decrease, Fold	FICI	MIC Decrease, Fold
*S. aureus* ATCC (MSSA)	0.375	8	0.75	2	1.25	1
*S. aureus* 18 (MSSA clinical isolate)	2.25	0.5	0.27	64	0.75	2
*S. aureus* 25 (MSSA clinical isolate)	0.27	64	1.25	1	0.37	8
*S. aureus* 26 (MSSA clinical isolate)	0.375	8	0.27	64	1.25	1
*S. aureus* 67 (MRSA clinical isolate)	1.25	1	0.26	128	1.25	1
*S. aureus* 68 (MRSA clinical isolate)	1.25	1	1.25	1	0.75	2
*S. aureus* 73 (MRSA clinical isolate)	0.25	1024	1.25	1	0.31	16
*S. mutans* (clinical isolate)	0.27	64	1.25	1	0.375	8
*S. sobrinus* (clinical isolate)	0.25	512	0.25	256	0.25	256
*S. salivarius* (clinical isolate)	1.25	1	0.25	256	0.25	256
*S. gordonii* (clinical isolate)	0.26	128	8.6	0.25	1.25	1

## Data Availability

The data presented in this study are available on request from the corresponding author.

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
