# Peer review of "The Antimicrobial Potential of the Hop (Humulus lupulus L.) Extract against Staphylococcus aureus and Oral Streptococci"

_pharmaceuticals, 2024, doi:10.3390/ph17020162_

Round 1
Reviewer 1 Report
Comments and Suggestions for Authors
In this study, the antimicrobial activity of extract from Humulus lupulus L against Staphylococcus aureus and oral Streptococci was investigated. The chemical components of the extract were analyzed by the thin-layer chromatography, UV-vis analysis and LC-MS analysis. In addition, the synergetic effect of the extract with various systemic antimicrobials was studied. Followings are some suggestions for the improvement of the manuscript.
1. It is difficult to understand “active pharmaceutical substances (APS)”, is it the extract from the plant?
2. The first sentence of abstract can be deleted.
3. The first and second paragraphs of introduction section can be combined into one paragraph, and the content can be greatly reduced.
4. The third paragraph of introduction section can be deleted.
5. The research progress about the synergetic effect of plant extracts with various systemic antimicrobials can be introduced.
6. The key innovation points and more detail research content of present work can be introduced in the last paragraph of introduction section.
7. Figure 1 and Figure 2 can be provided as supplementary material.
8. Typical LC-MS chromatogram of the extract should be provided, at least as supplementary material.
9. How to identify the chemical compounds by LC-MS should be described in detail, and related references should be cited.
10. Some of the content can be reduced. For example, the later part of section “2.4. Active pharmaceutical substance from H. lupulus kills bacterial cells in biofilm”.
11. The “synergetic effect” can be briefly summarized in the conclusion section.
12. The shortcomings of present work can be emphasized in the conclusion section.
13. There should be a space between the numeral and unit, the “m/z” should be in Italic font style, the “+” in “[M+]” should be in superscript.
Author Response
Dear Editor,
Please find enclosed the revised version of the paper “The antimicrobial potential of the Hop (Humulus lupulus L.) extract against Staphylococcus aureus and oral Streptococci” by Alyona Khaliullina, Alyona Kolesnikova, Leysan Khairullina, Olga Morgatskaya, Dilyara Shakirova, Sergey Patov, Polina Nekrasova, Mikhail Bogachev, Vladimir Kurkin, Elena Trizna and Airat Kayumov that we would like to resubmit to Journal Pharmaceuticals.
We thank Reviewers for their positive evaluation of our manuscript. We have revised the paper accordingly and have tried to incorporate all suggestions raised by reviewer.
In the following, we response to particular concerns raised by the Reviewers point by point.
Reviewer’s question:
It is difficult to understand “active pharmaceutical substances (APS)”, is it the extract from the plant?
Author’s response:
We agree with the reviewer that the term “active pharmaceutical substances (APS)” is difficult and not correct, therefore we changed it by the term “extract” throughout the text
Reviewer’s question:
The first sentence of abstract can be deleted.
Author’s response:
We agree with the reviewer, and removed the first sentence, as well as the whole abstract has been re-written.
Reviewer’s question:
The first and second paragraphs of introduction section can be combined into one paragraph, and the content can be greatly reduced.
The third paragraph of introduction section can be deleted.
Author’s response:
We thank the reviewer for valuable advices, and the introduction has been re-written and the content has been reduced
Reviewer’s question:
The research progress about the synergetic effect of plant extracts with various systemic antimicrobials can be introduced.
Author’s response:
Information has been added as suggested by the reviewer.
Reviewer’s question:
The key innovation points and more detail research content of present work can be introduced in the last paragraph of introduction section.
Author’s response:
The last paragraph has been extended as suggested by the reviewer.
Reviewer’s question:
Figure 1 and Figure 2 can be provided as supplementary material.
Author’s response:
We suggest to move the Figure 2 to supplementary material, while Figure 1 we would like to leave in the main text, if the reviewer would have no objections.
Reviewer’s question:
Typical LC-MS chromatogram of the extract should be provided, at least as supplementary material.
Author’s response:
The typical LC-MS chromatogram of the extract has been added to supplementary material as suggested by the reviewer.
Reviewer’s question:
How to identify the chemical compounds by LC-MS should be described in detail, and related references should be cited.
Author’s response:
The description of LC-MS has been extended as suggested by the reviewer.
Reviewer’s question:
Some of the content can be reduced. For example, the later part of section “2.4. Active pharmaceutical substance from H. lupulus kills bacterial cells in biofilm”.
Author’s response:
As suggested by the reviewer, we have shortened the text where appropriate
Reviewer’s question:
The “synergetic effect” can be briefly summarized in the conclusion section.
The shortcomings of present work can be emphasized in the conclusion section.
Author’s response:
The Conclusion section has been re-written and extended as suggested by the reviewer.
Reviewer’s question:
There should be a space between the numeral and unit, the “m/z” should be in Italic font style, the “+” in “[M+]” should be in superscript.
Author’s response:
Has been corrected
Dr. Alyona Khaliullina, Dr. Elena Trizna and Prof. Dr. Airat Kayumov, for all authors

Reviewer 2 Report
Comments and Suggestions for Authors
1. Title: The antimicrobial potential of the active pharmaceutical sub- 2 stance from HOP (Humulus lupulus L.) against Staphylococcus 3 aureus and oral Streptococci should be The antimicrobial potential of the active compounds from Humulus lupulus L. against Staphylococcus 3 aureus and oral Streptococci
2. The abstract should be re-written with a more detailed description of significant results
3. For what the arrows in Figure 1 refer to, it is not clear
4. Where the spots of authentic compounds in Figure 1
5. Table 1. LC/MS data of what?????
6. The authors should insert a table including of the inhibition % before MIC of antimicrobial assay
7. Several parts of the discussion section should be revised and rewritten
Comments on the Quality of English LanguageOK
Author Response
Dear Editor,
Please find enclosed the revised version of the paper “The antimicrobial potential of the Hop (Humulus lupulus L.) extract against Staphylococcus aureus and oral Streptococci” by Alyona Khaliullina, Alyona Kolesnikova, Leysan Khairullina, Olga Morgatskaya, Dilyara Shakirova, Sergey Patov, Polina Nekrasova, Mikhail Bogachev, Vladimir Kurkin, Elena Trizna and Airat Kayumov that we would like to resubmit to Journal Pharmaceuticals.
We thank Reviewers for their positive evaluation of our manuscript. We have revised the paper accordingly and have tried to incorporate all suggestions raised by reviewer.
In the following, we response to particular concerns raised by the Reviewers point by point.
Reviewer’s question:
Title: The antimicrobial potential of the active pharmaceutical sub- 2 stance from HOP (Humulus lupulus L.) against Staphylococcus 3 aureus and oral Streptococci should be The antimicrobial potential of the active compounds from Humulus lupulus L. against Staphylococcus 3 aureus and oral Streptococci
Author’s response:
We agree with the reviewers that the term “active pharmaceutical substances (APS)” is difficult and not correct, therefore we changed it by the term “extract” throughout the text. Consequently, we have changed the title of the paper as follow:
The antimicrobial potential of the Hop (Humulus lupulus L.) extract against Staphylococcus aureus and oral Streptococci
Reviewer’s question:
The abstract should be re-written with a more detailed description of significant results
Author’s response:
An abstract has been re-written as suggested by the reviewer.
Reviewer’s question:
For what the arrows in Figure 1 refer to, it is not clear
Where the spots of authentic compounds in Figure 1
Author’s response:
We apologize for the unclearity of the Figure. We rearranged arrows to show individual spots, and revised the figure caption by adding follow information: A pure luteolin (1), rutin (2), quercetin (3), and xanthohumol (10) available commercially were used as references and corresponding spots are shown by arrows.
Reviewer’s question:
Table 1. LC/MS data of what?????
Author’s response:
Due to technical mistake the title has been not full. We apologize for this inconvenience and provide the full title of the table
Reviewer’s question:
The authors should insert a table including of the inhibition % before MIC of antimicrobial assay
Author’s response:
The MIC values provided in Table 2 has been considered as a concentration of compound, providing the full repression of the visible growth, as judged by Alamar Blue viability test. Therefore we suggest to leave only Table 2
Reviewer’s question:
Several parts of the discussion section should be revised and rewritten
Author’s response:
The discussion has been revised as suggested by the reviewer.
Dr. Alyona Khaliullina, Dr. Elena Trizna and Prof. Dr. Airat Kayumov, for all authors

Reviewer 3 Report
Comments and Suggestions for Authors
Attach file

Comments on the Quality of English LanguageNo comments
Author Response
Dear Editor,
Please find enclosed the revised version of the paper “The antimicrobial potential of the Hop (Humulus lupulus L.) extract against Staphylococcus aureus and oral Streptococci” by Alyona Khaliullina, Alyona Kolesnikova, Leysan Khairullina, Olga Morgatskaya, Dilyara Shakirova, Sergey Patov, Polina Nekrasova, Mikhail Bogachev, Vladimir Kurkin, Elena Trizna and Airat Kayumov that we would like to resubmit to Journal Pharmaceuticals.
We thank Reviewers for their positive evaluation of our manuscript. We have revised the paper accordingly and have tried to incorporate all suggestions raised by reviewer.
In the following, we response to particular concerns raised by the Reviewers point by point.
Reviewer’s question:
Draw the structures of the main compounds: 2',4',6',4-tetrahydroxy-3'-geranylchalcone, cohumulone, colupulone, humulone and lupulone.
Author’s response:
The structures of compounds are shown in Table S1.
The Conclusion section has been re-written and extended as suggested by the reviewer.
Reviewer’s question:
- 12 line 494: …fluoride-resistant Streptococcus mutans in oral biofilm … change by …fluoride-resistant Streptococcus mutans in oral biofilm …
Author’s response:
Has been changed
Reviewer’s question:
Check all the scientific names: Put in italic
Author’s response:
Has been checked
Dr. Alyona Khaliullina, Dr. Elena Trizna and Prof. Dr. Airat Kayumov, for all authors

Round 2
Reviewer 1 Report
Comments and Suggestions for Authors
Most of the reviewer's comments have been addressed, and the manuscript has been revised accordinlgy. This manuscript may be accepted for publication after minor revisions: The detail descriptions about the identification of compounds by LC-MS analysis should be provided. It is difficult to identify a compound based on a molecular weight only.
Author Response
Dear Editor,
Please find enclosed the revised version of the paper “The antimicrobial potential of the Hop (Humulus lupulus L.) extract against Staphylococcus aureus and oral Streptococci” by Alyona Khaliullina, Alyona Kolesnikova, Leysan Khairullina, Olga Morgatskaya, Dilyara Shakirova, Sergey Patov, Polina Nekrasova, Mikhail Bogachev, Vladimir Kurkin, Elena Trizna and Airat Kayumov that we would like to resubmit to Journal Pharmaceuticals.
We thank Reviewers for their positive evaluation of our revised manuscript with minor corrections of the language and revision of LC/MS part as requested by the reviewer.
Reviewer’s question:
This manuscript may be accepted for publication after minor revisions: The detail descriptions about the identification of compounds by LC-MS analysis should be provided. It is difficult to identify a compound based on a molecular weight only.
Author’s response:
We agree with the Reviewer that this is difficult to identify a compound based on a molecular weight only. Therefore the identification has been performed by both retention time obtained HPLC and molecular ion mass (m/z) accordingly to literature data and two pure compounds (humulone и lupulone) as references for calibration. We are fully convinced that the present description of the procedure is enough for experts in LC/MS to repeat the measurements. The extended description of the LC/MS procedure is added to the new version of the manuscript.
Dr. Alyona Khaliullina, Dr. Elena Trizna and Prof. Dr. Airat Kayumov, for all authors
Reviewer 2 Report
Comments and Suggestions for Authors
The authors did the required revisions and the manuscript can be accepted in the present form
Comments on the Quality of English LanguageOk
Author Response
Dear Editor,
Please find enclosed the revised version of the paper “The antimicrobial potential of the Hop (Humulus lupulus L.) extract against Staphylococcus aureus and oral Streptococci” by Alyona Khaliullina, Alyona Kolesnikova, Leysan Khairullina, Olga Morgatskaya, Dilyara Shakirova, Sergey Patov, Polina Nekrasova, Mikhail Bogachev, Vladimir Kurkin, Elena Trizna and Airat Kayumov that we would like to resubmit to Journal Pharmaceuticals.
We thank Reviewers for their positive evaluation of our revised manuscript with minor corrections of the language and revision of LC/MS part as requested by the reviewer 1.
Dr. Alyona Khaliullina, Dr. Elena Trizna and Prof. Dr. Airat Kayumov, for all authors